# Structural and Functional Characterization of Orcokinin B-like Neuropeptides in the Cuttlefish (*Sepia officinalis*)

**DOI:** 10.3390/md20080505

**Published:** 2022-08-04

**Authors:** Maxime Endress, Céline Zatylny-Gaudin, Jérôme Leprince, Benjamin Lefranc, Erwan Corre, Gildas Le Corguillé, Benoît Bernay, Alexandre Leduc, Jimmy Rangama, Liza Mouret, Anne-Gaëlle Lafont, Arnaud Bondon, Joël Henry

**Affiliations:** 1Biologie des Organismes et Écosystèmes Aquatiques (BOREA) Muséum National D’histoire Naturelle, Sorbonne Université, Université de Caen-Normandie, Université des Antilles, CNRS-8067, IRD 207, Normandy University, F-75231 Paris, France; 2UNICAEN, Sorbonne Université, MNHN, UA, CNRS, IRD, Biologie des Organismes et Ecosystèmes Aquatiques (BOREA), F-14032 Caen, France; 3INSERM U1239, Rouen Normandy University, Neuroendocrine, Endocrine and Germinal Differentiation and Communication (NorDiC), PRIMACEN, F-76000 Rouen, France; 4CNRS, Sorbonne Université, FR2424 ABiMS, Station Biologique, F-29680 Roscoff, France; 5Normandy University, Post Genomic Platform PROTEOGEN, SF ICORE 4206, F-14032 Caen, France; 6Normandy University, CIMAP, UMP 6252 (CEA/CNRS/ENSICAEN/Normandy University), F-14032 Caen, France; 7Université Rennes, CNRS, ISCR-UMR 6226, COrInt, F-35000 Rennes, France

**Keywords:** cephalopods, *Sepia officinalis*, egg-laying, neuropeptides, neurohormones, biological activity, immunocytochemistry, NMR, 3D structure

## Abstract

The cuttlefish (*Sepia officinalis*) is a Cephalopod mollusk that lives in the English Channel and breeds in coastal spawning grounds in spring. A previous work showed that the control of egg-laying is monitored by different types of regulators, among which neuropeptides play a major role. They are involved in the integration of environmental cues, and participate in the transport of oocytes in the genital tract and in the secretion of capsular products. This study addresses a family of neuropeptides recently identified and suspected to be involved in the control of the reproduction processes. Detected by mass spectrometry and immunocytochemistry in the nerve endings of the accessory sex glands of the females and ovary, these neuropeptides are also identified in the hemolymph of egg-laying females demonstrating that they also have a hormone-like role. Released in the hemolymph by the sub-esophageal mass, a region that innervates the genital tract and the neurohemal area of the vena cava, in in vitro conditions these neuropeptides modulated oocyte transport and capsular secretion. Finally, in silico analyses indicated that these neuropeptides, initially called FLGamide, had extensive structural homology with orcokinin B, which motivated their name change.

## 1. Introduction

Cuttlefish (*Sepia officinalis*) are one of the main cephalopods fished on the English Channel coast. At the end of a life cycle of ca. 22 months, they breed between March and June in highly localized coastal spawning areas. Before the breeding season, the females perform gametogenesis in the autumn of the second year of their life cycle: an intense phase of cell multiplication takes place in the genital tract glands and in the ovaries, corresponding to previtellogenesis. This is followed in early winter by the biosynthesis of yolk proteins by the ovaries and of capsular products by the accessory glands of the genital tract, corresponding to vitellogenesis. The females are considered to be mature when ovulation starts, that is, when the mature oocytes are released into the genital coelom by the ovary. Egg-laying usually starts before mating, so the oocytes are stored in the genital coelom, and proximal oviduct contractions are inhibited by factors of ovarian origin, expressed and secreted by the oocytes [1,2,3,4].

The inhibition of the oviduct contractions is broken by mating, and then the oocytes resume their progress through the genital tract. This transport is made possible by the peristaltic contractions of the oviduct, which are modulated by neuropeptides, such as FMRFamide-related peptides (FaRPs), APGWamide-related peptides (APGWa-RPs), and sepiatocin [5,6,7], in association with ovarian peptides [1,8,9], and sexual pheromones [10]. During their transport and before their release into the mantle cavity, the oocytes are embedded in a first gelatinous capsule secreted by the oviduct gland (OvG), which forms the innermost layer of the future capsule. After their release into the mantle cavity, the partially encapsulated oocytes are embedded in a second envelope; this outermost layer is composed, among others, of proteins secreted by the main nidamental glands (MNGs), by bacteria, and of melanin from the ink pouch [11]. The secretion of the capsule, the inner and the outer layers, is regulated by the neuropeptides [6], and sex pheromones [10]. The role of the accessory nidamental glands in capsular secretion is suspected, but has never been demonstrated.

The encapsulated oocytes leave the mantle cavity through the siphon, probably carried by the gill current, and remain for ca. three minutes in a cavity formed by the female cuttlefish arms and buccal mass. The fertilization takes place there, with the spermatozoa stored in the copulatory pouch since mating. Because asynchronous gametogenesis results in a continuous production of mature oocytes, the majority of females are able to replenish their oocyte stock while renewing the capsular products. This mechanism results in multiple egg deposition periods interspersed with rest periods that make it difficult to assess fertility [12].

Although the neuropeptides are not the only regulators involved in the egg-laying regulation, they play a crucial role in the integration of the environmental cues, such as photoperiod, temperature, salinity, or depth.

In cuttlefish, the neuropeptides, such as APGWamide [7], FMRFamide [6], and sepiatocin [5], play a key role in oocyte transport and capsule secretion. In other mollusks, such as oyster (*Saccostrea glomerata*), several neuropeptides can induce spawning in sexually mature individuals: spawning hormone (ELH); GnRH; APGWamide; buccalines; crustacean cardioactive peptides (CCAPs); and LFRFamide [13]. In gastropods, the ELH appears to be the mainstay of the egg-laying regulation, particularly in hermaphrodite snails, such as *Aplysia* [14,15,16] and *Lymnea* [17,18,19], while in the gonochoric gastropods, several neuropeptides are also involved, in addition to ELH. In abalone (*Haliotis asinina*), several genes encoding APGWa, myomodulin, proctolines-like, FMRFa, schistosomine-like, insulin (MIP), and HGAP (haliotid growth-associated peptide) are expressed differentially in the males and females during the two-week spawning cycle [20]. Recent studies in cuttlefish have identified the entire neuropeptidome, which is composed of at least 38 distinct families [21]. Based on the patterns of expression and tissue localization, several families are strongly suspected to be involved in egg-laying control, particularly in the formation and release of encapsulated oocytes: APGWa, CCAPs, clionine, FLGamide, PTSP-like peptides, small cardioactive peptides (SCPs), MIP, myomodulin, sepiatocin, and SPamide. We chose to study the FLGamide family of neuropeptides, first identified by Zatylny-Gaudin et al. [21], and possibly related to arthropod orcokinins. We identified an incomplete N-terminal precursor, as well as two C-terminal ends from the assembly performed in the BioProject ID: PRJNA242869 (Figure 1A) and a new assembly realized from the SNC of juveniles. Six neuropeptides are cleaved (Figure 1B), as shown by the detection of these peptides by mass spectrometry in nerve endings, in the hemolymph, and in the CNS in the work of Zatylny-Gaudin et al. [21]. Figure 1C shows that the expression is only located in the CNS. The presence of three neuropeptides, with a C-terminal FLGamide end, initially led to the name “FLGamide neuropeptides”. These neuropeptides were therefore named based on their order of appearance on the precursors: FLGa 1 and FLGa 2 have a common PYY C-terminal end; FLGa 3, FLGa 4, and FLGa 5 are amidated in their C-terminal end; and FLGa 6 does not have any C-terminal amidation. The mature neuropeptides predicted by the in silico study were detected by mass spectrometry in the CNS, in the reproductive tract of the egg-laying females, and in the hemolymph. It is also the only family of neuropeptides detected so far in the ovary, which makes it possible to pose the hypothesis that the FLGamides could be involved in the first step of egg laying: ovulation. In *S. officinalis*, the mature ovary consists of a cluster of ovarian follicles at all of the stages of development. The mature follicles release the so-called smooth oocytes into the genital coelom (ovulation), where they are stored until spawning. The mechanism driving ovulation is still unknown. The release of the smooth oocytes could be induced by micro contractions at the level of the ovarian stroma, which is made up of connective fibers associated with many blood vessels and a few muscle fibers. The detection of the FLGamide in the ovarian stroma therefore suggests a potential hormonal action.

This study, therefore, aims to elucidate the possible role of the FLGamide in the regulation of the egg-laying mechanisms in cuttlefish. From this perspective, we studied the tissue expression of the FLGamides, using data from the transcriptome studies by Zatylny-Gaudin et al. [21]. We localized the neuropeptides by immunocytochemistry, using specific polyclonal antibodies, and by mass spectrometry, using a peptidomic approach. We evaluated the biological activity through a myotropic test performed on the different contractile organs of the female genital tract involved in egg-laying.

## 2. Results

### 2.1. Identification of FLGamide Precursors

The in silico analyses of the new transcriptome assembly identified two similar complete protein precursors (Figure 2), probably derived from alternative splicing. The first precursor, named FLGa A, encoded six neuropeptides; this precursor carried seven copies of the neuropeptide FLGa 3 and a single copy of the other neuropeptides. The second precursor, FLGa B, differed from FLGa A only by an additional copy of the neuropeptide FLGa 3.

### 2.2. Tissue Mapping by Mass Spectrometry

Mass spectrometry analyses were carried out on the distal part of the MNG, on the proximal oviduct, as well as on the most distal part of the oviduct (the flag) and the oviduct gland. The FLGa neuropeptides 1, 2, and 3 were detected in these three structures, as well as in the three main parts of the CNS (Appendix A). The other three neuropeptides predicted on the precursor were not detected.

### 2.3. Tissue Mapping by Immunocytochemistry

In the oviduct gland, the cells immunostained with the anti-FLGa antibody were detected in the secretory areas (Figure 3A). The nerve fibers labeled with the same antibody were also observed in the connective tissue that surrounds the gland. Similar observations were made with the anti-PYY antibody (Figure 3B). The oviduct contained endings labeled with the two antibodies, as well as cells with their cytoplasm labeled with the anti-FLGa antibody (Figure 3C,D). In the main nidamental gland, the bundles of labeled nerve fibers were abundantly detected in the distal portion of the gland, with both antibodies diluted at 1/500. In the ovarian stroma, the nerve endings labeled with anti-FLGa and anti-PYY antibodies (Figure 3E,F), diluted at 1/500, were detected. In the CNS, the anti-PYY and anti-FLGa antibodies were used at a 1/1000 dilution. The fibers and neurons labeled with both antibodies were observed in the paleovisceral lobe (Figure 4A,B).

### 2.4. Structure Determination

The complete NMR analysis of the peptide FLGa 1 was performed, in the presence of 150 mM of deuterated dodecylphosphocholine, using the sequential assignment strategy [22]. The complete chemical shift assignment is presented in Table 1. The pattern of the NOESY cross-peaks was characteristic of a helical structure of the peptide with numerous inter residue i–i + 3 connectivities. The structure was solved using AMBER software, and the superposition of the 18 lowest energy conformers of the peptide FLGa 1 in the DPC micelles, fitted on segment 4–14 backbone atoms, is presented in Figure 5. The summary of the refinement statistics is given in Table 2. An alpha helix is clearly defined from the residues 4 to 14; the first residues, as well as the C-terminal units, are unstructured.

### 2.5. Biological Activity

Two neuropeptides were used in the biological activity tests to represent each peptide type: peptide FLGa 1 for the neuropeptides with the PYY-type C-terminal end; and peptide FLGa 3 for the neuropeptides with the FLGamide-type C-terminal end. Increasing peptide concentrations were used to test the contractile activity of the ovarian stroma, distal oviduct, and main nidamental glands of the egg-laying and vitellogenetic females, and also of the penis and gills of the mature males. No activity was observed in the male organs or in the organs of the vitellogenetic females. Similarly, no activity was observed in the ovarian stroma, regardless of the cuttlefish maturity state.

Both of the neuropeptides had a dose-dependent myosuppressive activity on the mature distal oviduct: Figure 6A,B shows a sharp fall in the tonus at 10^−9^ M, then the tonus increases gradually.

The tests carried out on the mature main nidamental glands showed the heterogeneous biological activity of the two neuropeptides: at 10^−8^ M, FLGa 1 increased the amplitude of the contractions of the gland (Figure 7A). At 10^−6^ M, it also induced an increase in the tonus (Figure 7B). At 10^−8^ M, the neuropeptide FLGa 3 triggered a slight temporary increase in the tonus, followed by a slight temporary decrease in the amplitude of the contractions (Figure 7C). At 10^−6^ M, this same neuropeptide induced a marked decrease in the amplitude of the contractions (Figure 7D). After ca. fifteen minutes, the amplitude of the contractions returned to the basal level. In order to check the existence of a synergy between the FLGamide and the other neuropeptides, we conducted an experiment on the main nidamental gland (Figure 8). Firstly, 10^−6^ M FMRFamide was dripped into the vessel (Figure 8A). Then, the two neuropeptides were dripped simultaneously at 10^−6^ M (Figure 8B). The neuropeptide mixture triggered a sharp increase in the amplitude of the gland contractions. A similar, but more intense and lasting effect, was observed on this same gland with a simultaneous dripping of FMRFamide and FLGa 1 at 10^−6^ M (Figure 8C).

## 3. Discussion

The results of the present study provide insights into the role of a family of neuropeptides found in the whole *S. officinalis* female genital tract in the regulation of spawning.

The expression profile, determined from the 16 transcriptomes sequenced by Zatylny-Gaudin et al. [21] (BioProject PRJNA242869), made it possible to highlight that the transcripts were localized in the three main regions of the CNS (OL, SupEM, and SubEM). The SubEM is the region of the CNS that innervates the visceral mass and the genital tract, as well as the only neurohemal area described to date in cuttlefish [23,24,25,26]. Moreover, lower expression levels were observed in the glands involved in the secretion of the inner and outer capsules of the egg (Figure 1C).

The occurrence of the neuropeptide transcripts in these glands allows us to speculate about the mechanisms responsible for the rapid response induced by external stimuli. These mechanisms could be regulated by netrin-1, as described in *Aplysia californica* [27]. The binding of netrin-1 to the cytoplasmic domain of the netrin-1 receptor, called DCC (for Deleted Colorectal Cancer), increases translation of subcellular mRNAs located in dendrites or axons. The ultrastructure study revealed the occurrence of a rough endoplasmic reticulum, smooth reticulum and Golgi apparatus in the axonal compartment; therefore the mRNAs detected in axons could be translated very close to the target tissue [28].

Moreover, it is necessary to take into account the local expression by the isolated peripheral cells. The immunochemistry investigations in cuttlefish revealed immunostained nervous fibers and cells located in the MNG and OvG. The neuropeptides detected in these glands by mass spectrometry could, therefore, have three distinct, possibly complementary, origins: (1) they may be expressed in the cellular body of a central neuron of the SubEM and carried to the target organ via the axonal vesicles; (2) they may be expressed from the mRNAs delocalized in the terminal axons by nexin, as described in Aplysia [27]; (3) they may be expressed and secreted by the peripheral cells (neurons?) scattered across the glandular tissue, as observed in the present study.

The immunostained cells observed in the secretory tissue of the oviduct gland and MNG, along with a few labeled nerve fibers, differed from the adjacent secretory cells by a different morphology. This suggests that they were not glandular cells involved in the synthesis of capsule products. The endocrine cell dispersal at the peripheral level in the genital tract has already been described in different invertebrates, such as crustaceans, using cells immunostained with an anti-molluscan ELH antibody in the female gonad [29,30], or with a specific anti-ELH antibody in the gastropod, *Haliotis asinina* [31]. These cells may, therefore, be the source of local neuropeptide secretion in the target tissue, i.e., of paracrine regulation.

In the MNG, the labeled nerve fibers were mostly found at the most distal end of the gland, near the distal end of the main collecting duct, located in the main axis of the gland. Similar labeling has already been detected in this area with an anti-CCAP antibody [32]. This highly muscular canal contracts to discharge secretions into the mantle cavity when the oocytes pass through it. Therefore, our immunostaining results are in agreement with our mass spectrometry analyses.

In the CNS, immunostained neurons and fibers were detected in the paleovisceral and brachial lobes of the SubEM, as well as in the median lobe of the SupEM. These results confirm the mass spectrometry data from the SupEM and SubEM [21]. Moreover, they make it possible to specify the localization of the expression sites of the peptides in the lobes. The detection of the labeled neurons in the paleovisceral lobe is consistent with the presence of the immunostained nerve fibers in the accessory glands of the female genital tract, since this lobe innervates the viscera and the genital tract. Moreover, the paleovisceral lobe innervates the neurohemal area of the vena cava, and this is also in agreement with the detection of all of the mature neuropeptides by mass spectrometry in the neurohemal area and the hemolymph of the egg-laying females [21].

The biological activity tests revealed that FLGamide had a biological activity in the genital tract of the egg-laying females, but no activity was observed in the male genital organs or in the genital tract of the vitellogenetic females. The neuropeptides decreased the tonus of the distal oviduct in a dose-dependent manner, demonstrating their involvement in the oocyte transport through the modification of peristalsis. They can also be involved in blocking the oocytes to avoid their emission into the mantle cavity of unmated females. A similar activity from the serotonin of ovarian origin was already described in cuttlefish [33], and by SepCRPs [2,3]. At the level of the main nidamental gland, which secretes the outer capsule of the egg, the results of the myotropic tests were more heterogeneous. However, thanks to the synergy test carried out in this study, we highlighted that the FLGamide worked in association with FMRFamide, and increased its myoexcitatory effect. Nevertheless, the effects of the FLGamides alone showed that they are probably involved in the secretory mechanisms that release capsular products. As for the heterogeneity of the results, it was attributable to the fact that the cuttlefish were caught in the wild. It was impossible to tell whether the females we tested had ever laid eggs, once, twice, or more, or if they were senescent unless the first signs of loss of control of buoyancy and orientation were already visible. This means that, although the females we tested were all sexually mature, they were likely to be more or less advanced in their spawning period and, therefore, more or less receptive to stimulation. These parameters undoubtedly induced variations in the intensity of the responses observed during the myotropic tests.

On the other hand, it is important to take into account that, under physiological conditions, the target organs are subjected to the stimulation of a complex cocktail of regulatory peptides, ovarian peptides, neuropeptides, and even sexual pheromones for the organs of the genital system that bathe in the mantle cavity. The activity recorded when an isolated neuropeptide is applied on a target organ makes it possible to predict the presence of specific receptors. Yet, it does not in any way enable us to predict the characteristics of this activity, because the neuropeptide is part of a set that is lacking during the myotropic tests.

From the structural point of view, there is no defined structure of the peptide FLGa 1 in water, as generally encountered with a small peptide. However, in a micellar medium, an extended alpha helix is found after molecular modeling using NMR constraints. Such a peptide folding is consistent with a direct binding to the membrane receptor or membrane interaction, with the idea that the peptides are thought first to interact with the membrane and then migrate and bind specifically to their receptors [34,35].

The C-terminal PYY and FLGamide ends of the neuropeptides, predicted from mRNA and then confirmed by mass spectrometry, were quite unique according to currently available data. Nevertheless, it seems that this neuropeptide family presented important sequence homology with the neuropeptide families well described in protostomians, especially in arthropods. The multiple alignments revealed a strong conservation of the GGG domain, common to orcokinins B and C (Figure 9A–D). However, the hexapeptide domain DS(L/I)GGG, characteristic of FLGamide, appeared well preserved in orcokinin B. From a physiological viewpoint, in cockroach (*Blatella germanica*), orcokinin B is involved in the control of vitellogenesis and oocyte growth in adult females [36]. However, the localization of the cuttlefish FLGamides in nerve endings innervating the ovary suggests a similar role in the regulation of a vitellogenesis. This strong structural homology, associated with functional convergence, led us to rename this neuropeptide family So-orcokinin B. The first neuropeptide family had been annotated orcokinin B by Zatylny et al. [21]. However, given the high variability of the C-terminal domain of the DSI motif, it seemed to us more cogent to rename the former So-orcokonin A.

Moreover, we should point out an annotation error in *Octopus bimaculoïdes* and *Loligo pealeii*, in which orcokinin B is annotated “Feeding circuit activating neuropeptides” (FCANs). Figure 10A,B show a significant divergence between the consensus sequences of the insect FCANs and *Octopus* and *Loligo* pseudo-FCANs. In addition, So-orcokinin B has strong homology at the level of the primary sequences of *Octopus* and *Loligo* pseudo FCAN, especially with characteristic N-terminal elongations. Therefore, we believe that the *Octopus* and *Loligo* pseudo-FCANs are orcokinin B.

In summary, this article presents the first evidence that a neuropeptide family, initially called FLGamide and renamed So-orcokinin B, is expressed both in the CNS and peripheral cells associated with the glandular epithelium of cephalopods. So-orcokinin B may regulate the various steps of egg-laying, such as the oocyte transport and egg capsule secretion. As orcokinin B is both a neurohormone and neuromodulator in *S. officinalis*, we can suspect them to be involved in the regulation of the egg capsule biosynthesis. Further studies will allow us to check this hypothesis. On the other hand, the role played by So-orcokinin B in the ovary has to be elucidated. Although it is not involved in the release of oocytes into the genital coelom, as shown in this study, its occurrence in the nerve endings of the ovarian stroma suggests a possible involvement in the regulation of vitellogenesis, as described in insects. Finally, this study allowed us to correct the annotations of *S. officinalis* FLGamide, as well as *O. bimaculoides* and *L. pealeii* FCANs, by newly annotating them as orcokinin B.

## 4. Materials and Methods

### 4.1. Animals and Tissue Collection

All of the mature cuttlefish were between 1 kg and 1.5 kg with a dorsal length of 18 to 25 cm. They were trapped in the Bay of Seine from April to June 2015, 2016, and 2017. They were maintained in 1000-L outflow tanks at 15 ± 1 °C at the Marine Station of Luc-sur-Mer (University of Caen-Normandy, France) under a natural photoperiod. The organs were dissected from animals anesthetized with 3% ethanol [37] and then immediately frozen in liquid nitrogen, or stored in synthetic seawater (Reef crystal^®^, Instant Ocean, Blacksburg, VA, USA) containing 1 mM glucose and maintained at 18 °C, or fixed in Davidson solution. All of the applicable guidelines for the care and use of animals were followed. The procedures were approved of by the regional Ethical Committee (Comité d’Ethique Normandie en Matière d’Expérimentation Animale, CENOMEXA; agreement number 54).

### 4.2. In Silico Analyses

The transcriptome sequencing, assembly and annotation are described by Zatylny-Gaudin et al. [21]. The expression was quantified using fragments per kilobase of exon per million fragments mapped (FPKM) values from the 16 tissue-specific transcriptomes. The FPKM values of a given transcript from several tissues were compared to establish an expression pattern. In this case, the FPKM values represented the pooled expression of five animals used to determine each transcriptome.

The multiple sequence alignments of FLGamide, orcokinins, feeding-circuit peptides, FLGamide precursors from mollusks and arthropods were performed using CLC Main Workbench 6.7.1 and SIM-Alignment Tool for protein sequences (Expasy). The *Loligo pealei* transcriptomic data were retrieved from the website http://ivory.idyll.org/blog/2014-loligo-transcriptome-data.html (accessed on 17 February 2014), where the *L. pealeii* genome and five transcriptomes are available, as well as a public Blast server. The protein precursors of FLGamide came from a first assembly by Zatylny-Gaudin et al. [21] (BioProject PRJNA242869, https://zenodo.org/record/3647818#.YI-8emYzZqs, accessed on 11 December 2019) and a second supplemental one corresponding to the transcriptomes of juvenile CNS (296520 transcripts).

### 4.3. Tissue Mapping by Mass Spectrometry

#### 4.3.1. Extraction

Each tissue extraction was performed from three animals. The tissues were crushed in liquid nitrogen and extracted for 30 min with cold methanol/water/acetic acid (90/9/1), adjusted to 50 mM dithiothreitol (DTT). One gram of tissue was used for 10 mL of extraction buffer. Each extract was centrifuged 20 min at 20,000× *g* at 4 °C, and then the supernatant was evaporated in a speed vac. The dry pellets were resuspended in 0.1% trifluoroacetic acid (TFA) and concentrated on C18 Sep-Pak cartridges (Waters).

#### 4.3.2. NanoLC-MALDI-TOF/TOF Analysis

##### Sample Preparation for Mass Spectrometry Analysis

The concentrated and desalted peptide pellets were reduced with 100 mM DTT at 55 °C for 60 min, alkylated with 50 mM iodoacetamide at 55 °C for 45 min, and then concentrated and desalted on C18 OMIX-tips (10 μL, AGILENT). The chromatography step was performed on a nano-LC system (Prominence, Shimadzu, Kyoto, Japan). The peptides were concentrated on a Zorbax 5 × 0.3 mm, 5µm C18 precolumn (Agilent), and separated on a Zorbax 150 × 0.075 mm, 3.5µm C18 column (Agilent). The mobile phases consisted of 0.1% TFA in 99.9% water (*v*/*v*) (A) and 0.1% TFA in 99.9% acetonitrile (ACN) (*v*/*v*) (B). The nanoflow rate was set at 300 nL/min, and the gradient profile was as follows: constant 2% B for 5 min; from 2 to 5% B in 1 min; from 5 to 32% B in 144 min; from 32 to 70% B in 10 min; from 70 to 90% B in 5 min; and back to 2% B in 10 min. The 300 nL/min volume of the peptide solution was mixed with 1.2 µL/min volumes of solutions of 5 mg/mL of an α-cyano-4-hydroxycinnamic acid (CHCA) matrix prepared in a diluent solution of 50% ACN containing 0.1% TFA. Twenty-second fractions were spotted by an AccuSpot spotter (Shimadzu) on stainless steel Opti-TOF™ 384 targets.

##### Mass Spectrometry Analysis

The MS experiments were carried out on an AB Sciex 5800 proteomics analyzer equipped with TOF TOF ion optics and OptiBeam™ (OptiBeam Antenna Technologies, Mühlacker, Germany) on-axis laser irradiation with 1000 Hz repetition rate. The system was calibrated before analysis with a mixture of des-Arg-bradykinin, angiotensin I, Glu1-fibrinopeptide B, ACTH (18–39) and ACTH (7–38), and mass precision was better than 50 ppm in reflectron mode. A laser intensity of 3400 was typically employed for ionizing. The MS spectra were acquired in the positive reflector mode by summarizing 1000 single spectra (5 × 200) in the 700 to 4000 Da mass range. The MS/MS spectra from the twenty most intense ions were acquired in the positive MS/MS reflectron mode by summarizing a maximum of 2500 single spectra (10 × 250) with a laser intensity of 4300. For the tandem MS experiments, the acceleration voltage was 1 kV, and air was used as the collision gas. Gas pressure medium was selected as the setting.

##### Peptide Sequencing and Protein Precursor Identification

The fragmentation pattern based on the occurrence of y, b, and a ions was used to determine peptide sequences. The database searching was performed using the Mascot 2.5.1 program (Matrix Science, Boston, MA, USA). A database corresponding to a homemade *S. officinalis* transcript database (including 356,644 entries) was used (BioProject PRJNA242869).

The variable modifications allowed were as follows: methionine oxidation and dioxidation; C-terminal amidation; and N-terminal pyroglutamate. “No enzyme” was selected. Mass accuracy was set to 200 ppm and 0.6 Da for the MS and MS/MS modes, respectively.

### 4.4. Tissue Mapping by Immunocytochemistry

The female CNS, the distal end of the oviduct, the OvG, MNGs, and the ovarian stroma were freshly dissected and placed in Davidson solution (30% filtered sea water, 30% ethanol 95%, 20% formalin 37%, 10% glycerol, and 10% acetic acid for 24 to 48 h at 4 °C. The tissues were dehydrated in several baths containing increasing ethanol contents. The tissues were totally dehydrated by a bath of butanol before paraffin inclusion. The 5-µm sections were deparaffined with Roti^®^-Histol (Carl Roth^®^, Carl Roth, Karlsruhe, Germany), and incubated in H_2_O_2_-methanol 3% for 10 min at room temperature. The rehydration was performed in several baths of decreasing ethanol contents, Tris buffer (Tris-HCl 100 mM, pH 7.4), and TT buffer (Tris buffer, 0.5% TritonX-100). The neutralization of non-specific sites was performed by incubation in bovine serum albumin (BSA) 3% in TT buffer. Diluted (1:500 or 1:1000) primary antibodies (Rabbit, GeneCust^®^ (GeneCust, Boynes, France), directed against CVFDTLGGGHVPYY and CFDSLGGGSFLG-amidation) in the TT buffer were applied on the sections and incubated overnight at 4 °C. Conjugated anti-rabbit antibody peroxidase diluted 1:500 in TT buffer (Invitrogen™ A16029, Thermo Fisher Scientific, Waltham, MA, USA) was applied on the sections and incubated for 1 h at room temperature. The cells and nerve fibers containing immunolabelled material were visualized using diaminobenzidine (SIGMA*FAST*™ (Sigma-Aldrich, Saint Louis, MO, USA) 3,3′-diaminobenzidine tablets) as a chromogen. The slides were finally counterstained with hematoxylin for 30 s. The primary antibody was preabsorbed with 20 µg/mL of antigen as a negative control for specificity. Alternatively, the primary antibody was omitted.

### 4.5. Peptide Synthesis

The peptides were synthesized by solid phase methodology on a 433A Applied Biosystems peptide synthesizer at a 0.1-mmol scale, using the standard procedures, as previously described [38]. All of the Fmoc-amino acids (1 mmol, 10 eq.) were coupled, either on a Fmoc-Gly-HMP, a Fmoc-Leu-HMP, a Fmoc-Tyr(*t*Bu)-HMP, a Fmoc-Val-HMP, or a Rink amide resin, by in situ activation with HBTU (0.9 mmol, 9 eq.) and DIEA (2.5 mmol, 25 eq.) in NMP including a capping step (Ac_2_O), before Fmoc removal (piperidine 20 % in NMP). After completion of the chain assembly, the peptides were cleaved from the resin by adding 10 mL of a TFA/TIS/H_2_O (9.5:0.25:0.25) mixture for 120 min at room temperature, as previously described [39]. The crude peptides were precipitated by adding TBME, centrifuged (4500 rpm), washed three times in TBME, and freeze-dried. The synthetic peptides were purified by reversed-phase HPLC on a 2.2 × 25 cm Vydac 218TP1022 C_18_ column using a linear gradient (10 to 50% in 45 min) of ACN/TFA (99.9:0.1), at a flow rate of 10 mL/min. The purified peptides were then characterized by MALDI-TOF mass spectrometry on an UltrafleXtreme (Bruker, Strasbourg, France) in the reflector mode, using α-cyano-4-hydroxycinnamic acid as a matrix. The analytical HPLC, performed on a 0.46 × 25 cm Vydac 218TP54 C_18_ column, indicated that the purity of all of the peptides was >99.9%.

### 4.6. NMR and Structure Determination

#### 4.6.1. NMR Measurements

The samples for NMR contained 1–3 mM of peptide FLGa 1 dissolved in water or in the presence of DPC micelles (150 mM). All of the spectra were recorded on a Bruker Avance 500 spectrometer, equipped with a 5 mm TCI cryoprobe (^1^H, ^13^C, ^15^N). The homonuclear 2-D spectra DQF-COSY, TOCSY, and ROESY or NOESY were typically recorded using standard Bruker sequences in the phase-sensitive mode, using the States-TPPI method. The NOESY or ROESY spectra were acquired with eight scans, eight spectra were summed leading to less t1 noise, as recently reported [40]. The typical spectra were acquired, using the matrices of 4096 × 320–600 zero filled in F1 to 2 K × 1 K after apodization with shifted sine-square multiplication in both of the domains. The spectra were processed with Topspin software (Bruker).

#### 4.6.2. Structure Calculations

The ^1^H chemical shifts were assigned according to classical sequential assignment procedure. The NOESY cross-peaks were integrated and assigned within the CcpNmr [41] and NMRView software [42]. The volumes of the NOESY peaks between the methylene pair protons were used as the reference of 1.8 Å. The structure calculations were performed with AMBER 17 [43] in two stages: cooking, and simulated annealing in an explicit solvent. The cooking stage was performed at 1000 K to generate 100 initial random structures. The simulated annealing calculations were carried out during 20 ps (20,000 steps, 1 fs long). First, the temperature was raised quickly and was maintained at 1000 K for the first 5000 steps, then the system was cooled gradually from 1000 K to 100 K from step 5001 to 18,000, and finally the temperature was brought to 0 K during the 2000 remaining steps. For the 3000 first steps, the force constant of the distance restraints was increased gradually from 2.0 kcal.mol^−1^.Å to 20 kcal.mol^−1^.Å. For the remaining simulations of the simulation (step 3001 to 20,000), the force constant was kept at 20 kcal.mol^−1^.Å. The 20 lowest energy structures with no violations >0.3 Å were considered as representative of the compound structure. All of the dihedral angles ϕ and ψ belonged to the allowed regions of the Ramachandran plot. The representation and quantitative analyses were carried out using MOLMOL [44].

### 4.7. Myotropic Bioassay

The myotropic bioassay was performed using the different contractile organs involved in egg-laying: the MNGs; the distal oviduct; the proximal oviduct; and the ovarian stroma. Each organ was suspended on a dynamometer (Dynamometer UF1, Pioden Controls Ltd., Kent, UK) in a muscle chamber, with a nylon thread (0.12-mm diameter). The signal was amplified by an amplifier (SGA 920201-01, Bionics Instrument, Tokyo, Japan) and the contractions were displayed on a printer (L200E, Linseis, Selb, Germany) or with a Serial Arduino plotter_1.1. This second imaging method is a homemade digital voltmeter, based on Arduino technology (https://www.arduino.cc, accessed on 15 November 2020), coupled to the Qt application programming interface (https://www.qt.io, accessed on 31 May 2022). The muscle chamber was perfused at a flow rate of 0.5 mL.min^−1^ with perfusion solution and maintained at 18 °C. Increasing concentrations of synthetic peptides were injected into the perfusing flow, using a three-way valve to avoid mechanical stress. The flow of the samples into the muscle chamber was traced by adding phenol red. Before each injection of synthetic peptide, the injection of perfusion liquid was performed, to which the phenol red was added.

## Figures and Tables

**Figure 1 marinedrugs-20-00505-f001:**
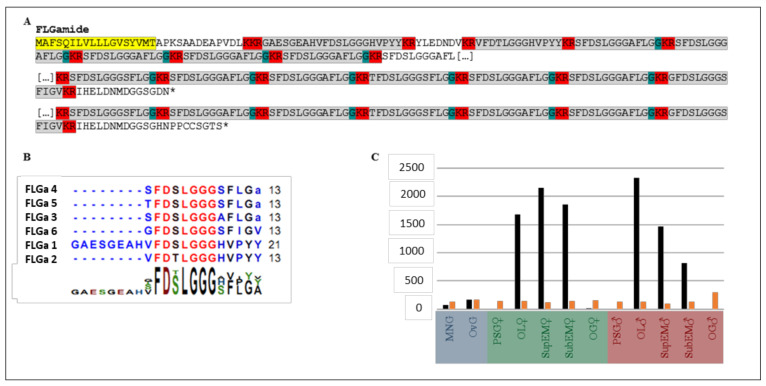
FLGamide family in *Sepia officinalis*: protein precursors, neuropeptides, and expression profiles (according to Zatylny-Gaudin et al., 2016 [21]). (**A**): Partial sequences, in amino acids, of FLGamide protein precursors. Highlighted in grey: neuropeptides predicted; in yellow: signal sequence; in red: tri- or dibasic cleavage sites; in blue-green: amidation in C-terminal extremity; * translation of stop codon (**B**): Predicted neuropeptides from sequences identified, alignments were performed with Clustal Omega; (**C**): Expression expressed in fpkm (fragments per kilo base per million mapped reads), of FLGamide precursor in egg-laying female and mature male. Orange bars for elongation protein 3 as a reference gene; ANG: Accessory Nidamental Gland; MNG: Main Nidamental Gland; OvG: Oviduct Gland; PSG: Posterior Salivary Gland; OL: Optic Lobe; SupEM: Supraesophageal Mass; SubEM: Subesophageal Mass; OG: Optic Gland. Highlighted in blue: female genital tract; in green: CNS female; in red: CNS male.

**Figure 2 marinedrugs-20-00505-f002:**
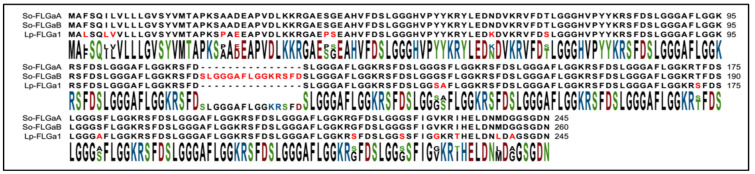
Sequence alignment of FLGamide protein precursors in *Sepia officinalis* and *Loligo pealeii*.

**Figure 3 marinedrugs-20-00505-f003:**
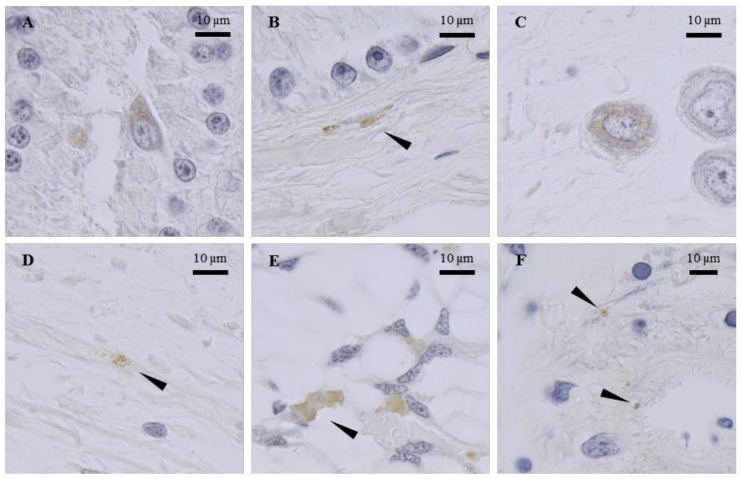
Tissue mapping by immunocytochemistry of PYY and FLGa neuropeptides. Transversal sections of genital apparatus of mature female cuttlefish. (**A**): Presence of cells with immunostained cytoplasm in the OvG with FLGa antibodies; (**B**): Presence of immunostained end fibers in the OvG with PYY antibodies; (**C**): Presence of cells with immunostained cytoplasm in the wall of the oviduct with FLGa antibodies; (**D**): Presence of immunostained end fibers in the wall of the oviduct with PYY antibodies; (**E**): Presence of immunostained fiber bundles in the distal end of the MNG with PYY; (**F**): Presence of immunostained end fibers in the ovarian stroma with PYY antibodies. Arrowheads indicate the immunostained end fibers.

**Figure 4 marinedrugs-20-00505-f004:**
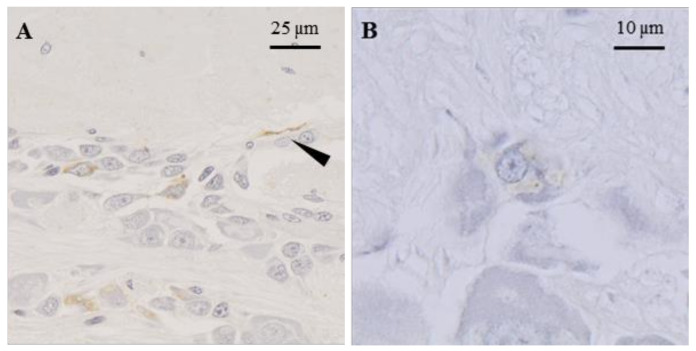
Immunocytochemical detection of neuropeptides of the FLGamide family in the SubEM of the paleovisceral lobe in female cuttlefish. (**A**): A group of neurons and a nerve fiber immunostained with the PYY antibodies; (**B**): A neuron with immunostained cytoplasm with FLGa antibodies. Arrowhead indicates immunostained nerve fiber.

**Figure 5 marinedrugs-20-00505-f005:**
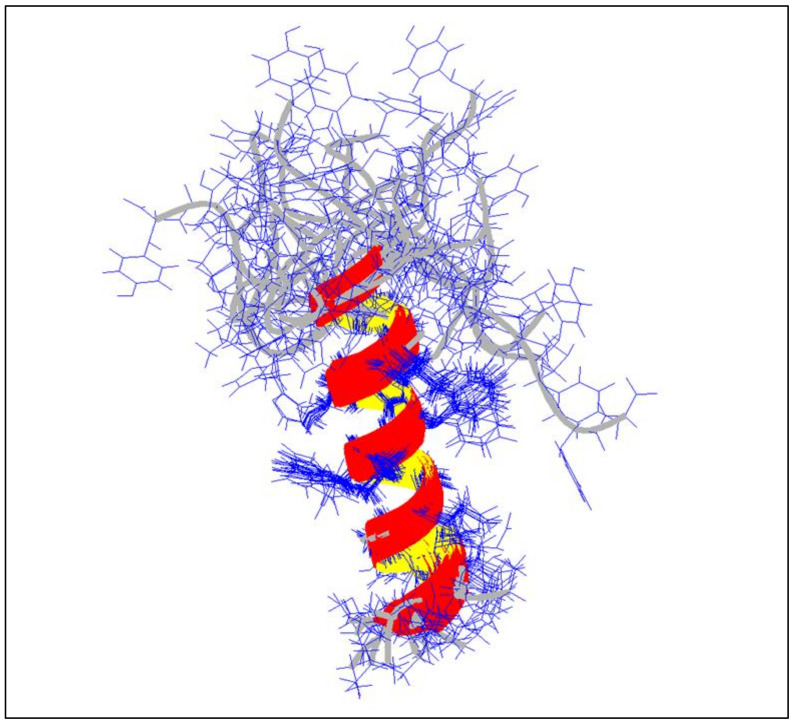
Superimposition of the 18 lowest energy conformers of the peptide FLGa 1 in DPC micelles, fitted on segment 4–14 backbone atoms (rmsd: 0.260).

**Figure 6 marinedrugs-20-00505-f006:**
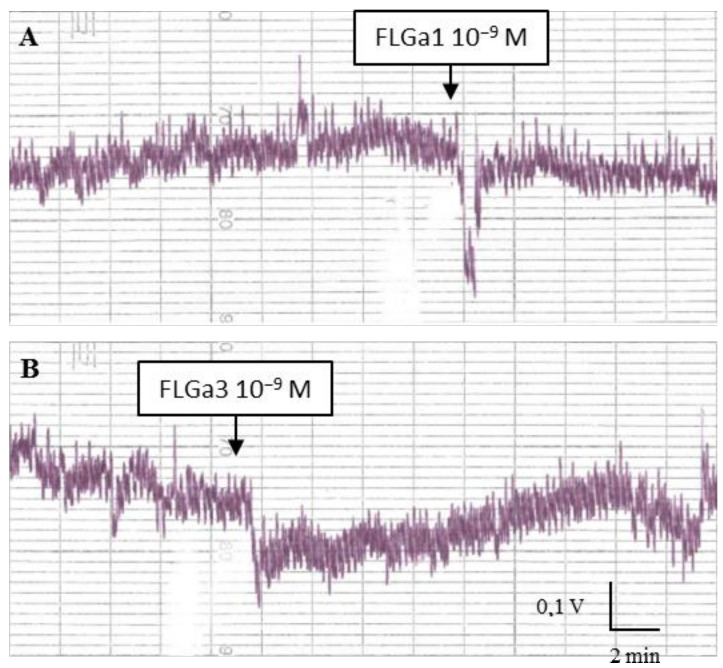
Biological activity of neuropeptides (**A**): FLGa 1 and (**B**): FLGa 3 applied to the distal oviduct of a mature cuttlefish. The two neuropeptides triggered a reversible fall of the oviduct tonus.

**Figure 7 marinedrugs-20-00505-f007:**
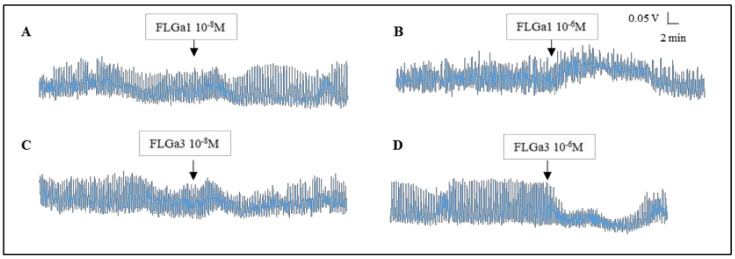
Biological activity of the neuropeptides FLGa 1 (**A**,**B**) and FLGa 3 (**C**,**D**) on the MNG of a mature cuttlefish. FLGa 1 tended to have a myoexcitatory effect, whereas FLGa 3 tended to have a myosuppressive effect.

**Figure 8 marinedrugs-20-00505-f008:**
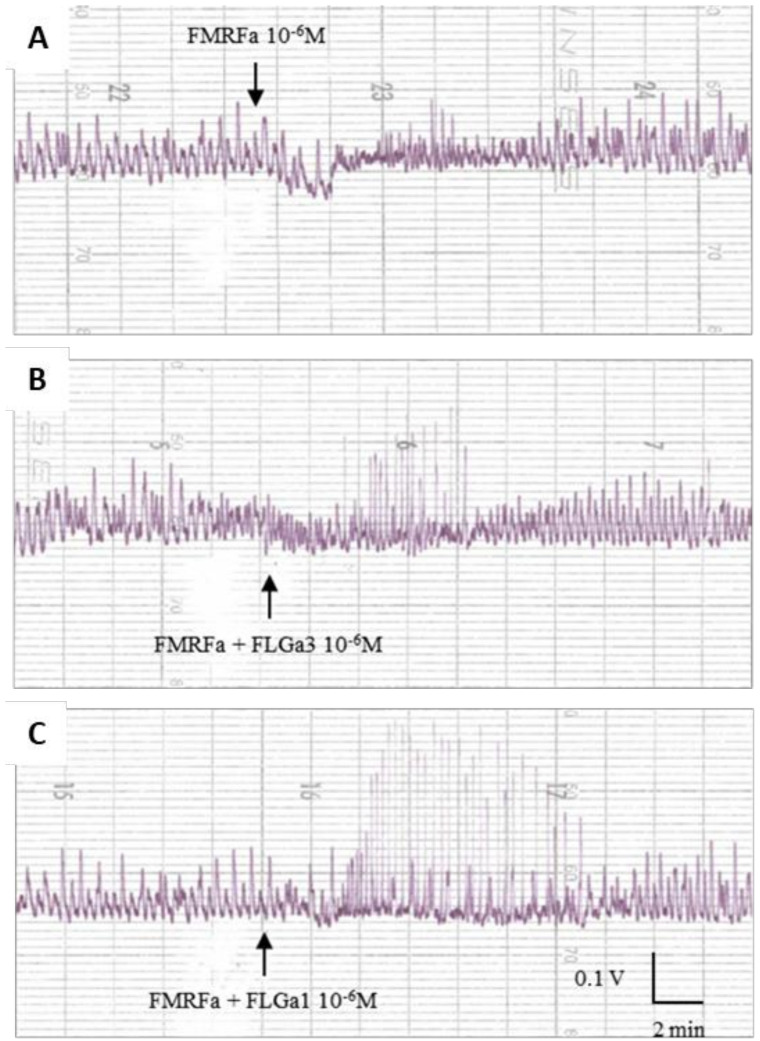
Synergistic activity of FLGamide with FMRFa, biological activity on MNG of mature cuttlefish. (**A**): FMRFa alone; (**B**): potentiation of FMRFa by FLGa3; (**C**): potentiation of FMRFa by FLGa1. Neuropeptides of the FLGamide family acted in synergy with FMRFamide on MNG, triggering a temporary myoexcitatory effect.

**Figure 9 marinedrugs-20-00505-f009:**
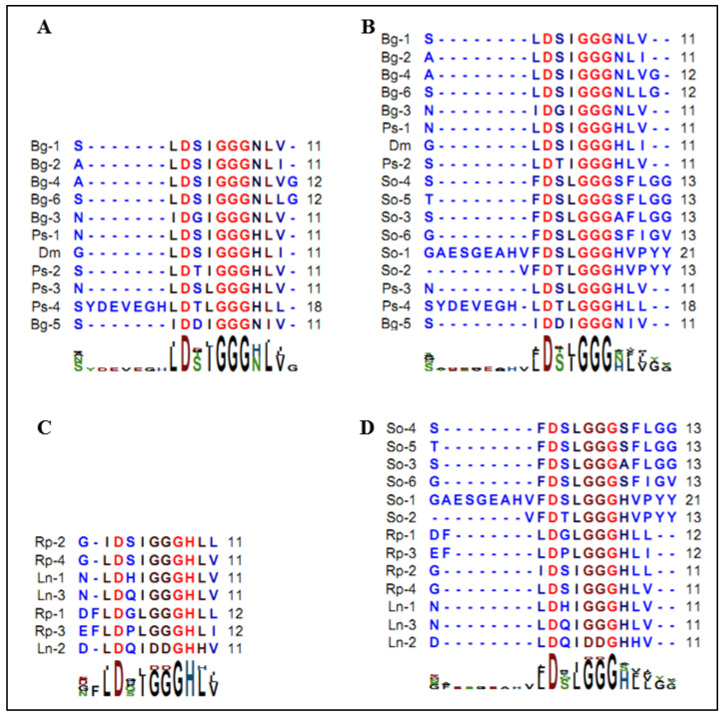
(**A**): Pattern of orcokinin B in insects. Bg: *Blatella germanica* (AKR13996.1). Ps: *Plautia stali* (BAV788822.1); Dm: *Drosophila melanogaster* (NP_002261160.1); (**B**): Multiple sequence alignments of orcokinin B of Bg, Ps, Dm, and FLGamide of *S. officinalis*; (**C**): Pattern of orcokinin C in insects; Rp: *Rhodnius prolixus* (AGW15565.1); Ln: *Lasius niger* (KMQ93900.1); (**D**): Multiple sequence alignments of orcokinin C of Rp, Ln, and FLGamide of *S. officinalis*.

**Figure 10 marinedrugs-20-00505-f010:**
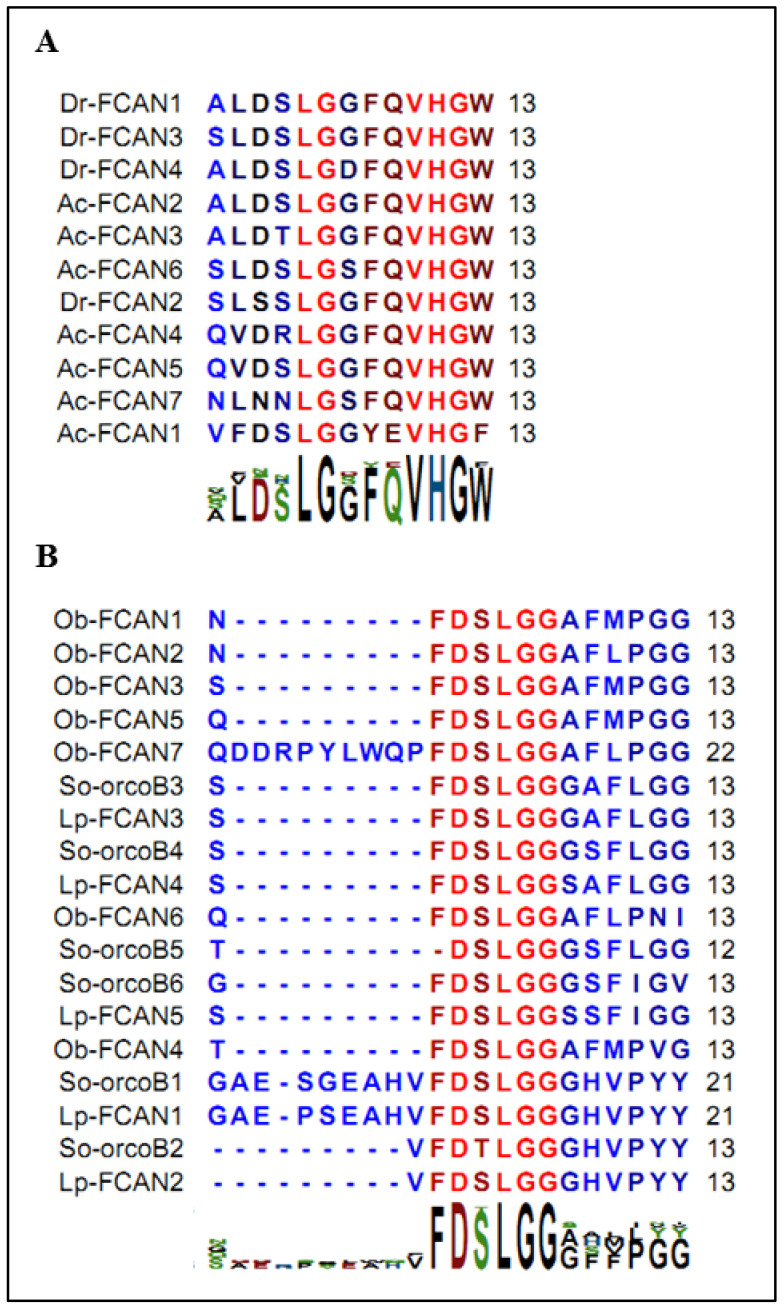
(**A**): Pattern of feeding circuit activating neuropeptides (FCANs) in insects. Bg: *Blatella germanica* (AKR13996.1); Ps: *Plautia stali* (BAV788822.1); Dm: *Drosophila melanogaster* (NP_002261160.1); (**B**): Alignment of pseudo-feeding circuit activating neuropeptides of *Octopus bimaculoides* (XP_014768639.1), *Loligo pealeii* with So-orcokinin B of *S. officinalis*.

**Table 1 marinedrugs-20-00505-t001:** NMR data and refinement statistics for the 18 best solution structures of the peptide FLGa 1 bound to DPC micelles.

Conformational Restraints
Distance constraints: 119
Restraints violations
>0.3 Å: 0
Mean global r.m.s.d. (Å)
Heavy atoms (residues 4–14): 0.26
Backbone atoms (residues 4–14): 0.746

**Table 2 marinedrugs-20-00505-t002:** NMR data and refinement statistics for the 18 best solution structures of the peptide FLGa 1 bound to DPC micelles. Number designed the amino acid position.

Residue	H^N^	HA	HB2	HB3	HG2	HG3	HD1	HD2	Others
Gly^1^	-	3.92/3.92							
Ala^2^	8.74	4.38	1.44						
Glu^3^	8.58	4.40	2.03	2.14					
Ser^4^	8.34	4.46	3.94	4.03					
Gly^5^	8.68	4.03/4.03							
Glu^6^	8.44	4.19	2.07	2.07	2.43	2.43			
Ala^7^	8.24	4.18	1.44						
His^8^	8.24	4.64	3.24	3.32			7.34		HE1 8.66
Val^9^	8.14	3.81	2.08		0.78	0.94			
Phe^10^	8.08	4.44	3.13	3.25			7.28	7.28	HE1/2 7.28 HZ 7.17
Asp^11^	8.32	4.55	2.91	2.91					
Ser^12^	8.04	4.41	3.92	3.97					
Leu^13^	7.84	4.36	1.81	1.81	1.64		0.91	0.91	
Gly^14^	8.13	3.99	3.99						
Gly^15^	8.22	3.93	3.93						
Gly^16^	8.32	3.91	3.91						
His^17^	8.12	3.18	3.30				7.31		HE1 8.63
Val^18^	8.27	4.35	2.13		1.02	1.02			
Pro^19^	-	4.41	1.82	2.17	1.92	1.98	3.57	3.89	
Tyr^20^	7.80	4.38	3.02	3.02			7.06	7.06	HE1/2 6.82
Tyr^21^	7.60	4.49	2.93	3.02			7.06	7.06	HE1/2 6.82

## Data Availability

Bioproject PRJNA242869 and https://zenodo.org/record/3647818# (accessed on 11 December 2019). YI-8emYzZqs.

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
