# Peer review of "Structural and Functional Characterization of Orcokinin B-like Neuropeptides in the Cuttlefish (Sepia officinalis)"

_marinedrugs, 2022, doi:10.3390/md20080505_

Round 1
Reviewer 1 Report
This article presents the first evidence that a neuropeptide family initially called FLGamide and renamed So-orcokinin B and suspect them to be involved in the regulation of egg capsule biosynthesis. The methods of mass spectrometry, tissue mapping by immunology and Myotropic bioassay used in the experiment are all advanced. The experimental results are clearly expressed, but there are some problems in the pictures of the experimental results, such as unclear and color difference. The discussion and methods can be appropriately simplified.The overall content of the article is substantial, and the neuropeptide studied is also a hot topic.To sum up, the article is innovative, rigorous in experimental methods and clear in writing logic. The article may be accepted after minor modification
Minor questions:
1.Line 67: “A”changed to "the"
2.Line99: delete “the figure”
3.Line119: “therefor”changed to ",therefor,"
4.Line212: delete “a”
5.Line294: “sentence,” delete “,”
6.Line308: “media”changed to "media,"
7.Line314: “current”changed to "currently"
8.Line317: delete “a”
9.Line335: the first“first”changed to "the first"
10.The explanation of the mechanism in the introduction needs to be simplified.
11.In Figure.4, The hue difference between A and B is too big
12.Figure3 and 4 need to be improved in clarity.
13.What are the specifications of the cuttlefish, such as age and weight?
14.Where does the tissue used for transcriptome sequencing and tissue mapping by mass spectrometry come from?
15.Will the peptide synthesis part of the methods section be too detailed?
Author Response
Dear reviewer,
we thank you for your comments which helped us to improve our article.
Sincerely yours
Joël Henry
Concerning
1.Line 67: “A” changed to "the"
Change made in the text.
2.Line99: delete “the figure”
Change made in the text.
3.Line119: “therefor” changed to ", therefor,"
Change made in the text.
4.Line212: delete “a”
Change made in the text.
5.Line294: “sentence,” delete “,”
Change made in the text.
6.Line308: “media”changed to "media,"
Change made in the text.
7.Line314: “current”changed to "currently"
Change made in the text.
8.Line317: delete “a”
Change made in the text.
9.Line335: the first“first”changed to "the first"
Change made in the text.
10.The explanation of the mechanism in the introduction needs to be simplified.
The Sepia officinalis cuttlefish is not a conventional model, so in my opinion it is necessary to give certain information to allow readers who are not specialists in invertebrates and cephalopods in particular to have a good understanding of the model studied.
11.In Figure.4, The hue difference between A and B is too big
I have not the possibility to correct it.
12.Figure3 and 4 need to be improved in clarity.
I don’t understand the request.
13.What are the specifications of the cuttlefish, such as age and weight?
Add in text.
14.Where does the tissue used for transcriptome sequencing and tissue mapping by mass spectrometry come from?
For the transcriptome, the procedure is described in Zatylny et al, (2016). For the mass spectrometry analyses, it is indicated on line 362 that the animals come from the Bay of Seine and were captured between April and June in 2015, 2016 and 2017.
15.Will the peptide synthesis part of the methods section be too detailed?
Modified in the text.
Reviewer 2 Report
The manuscript of Joël Henry focuses on ‘Structural and functional characterization of orcokinin B-like 2 neuropeptides in the cuttlefish (Sepia officinalis)’. The authors aim to elucidate the possible role of FLGamide in the regulation of egg-laying mechanisms in Sepia officinalis. Tissue expression of FLGamides was studied only using data from transcriptome studies, the neuropeptides were localized by immunocytochemistry, and the peptide structure was determinated by mass spectrometry, and the biological activity was evaluated through a myotropic test performed on the different contractile organs involved in egg-laying. The paper results interesting given new insights about the neuropeptides in cuttlefish. However, major revisions are required before acceptance for publication.
1) Rewrite the section of Introduction. In this section, the authors keyed point on the process of egg laying and mating of the cuttlefish, FLGamides or orcokinin B-like neuropeptides were not presented adequately. Please add more introduction about your aimed peptides.
2) Section 2.1, FLGamide precursors were only identified by transcriptome information and this is not enough to confirm your results and followed experiments. The precursors should be cloned and identified by molecular experiments.
3) Section 2.3 Tissue mapping by immunocytochemistry was performed. But in Figure 3 and figure 4, the results showed that low expressions of FLGamides and PYY were observed. How to explain the neuropeptides function at low expression level?
4) As we know than several mature peptides were predicted. Some did not show the functions what the authors expected. Whether the authors tried to use combined peptides to carry out the experiment? The results might be different. For example, FlGa1+FlGa3?
5) If possible, I suggest the authors to carry out the experiments with two antibodies at the same treatment.
6) Line 199, how can the authors get the results that both neuropeptides had a dose -dependent activity on the mature distal oviduct?
7) Section 2.5, biological activity, the positive or negative control should be added.
8) Line 360: Mention the body weight and shell length with mean ± SD.
9) Line 40, 109, 256…., Latin names should be italic.
10) Line 480, revise the format of the title.
11) Line 178, ‘are’ should be ‘is’
12) References: some of them lack of doi information and the Latin names in the references should also be ‘italic’.
Author Response
Dear reviewer,
we thank you for your comments which helped us to improve our article.
Sincerely yours
Joël Henry
Concerning
- Rewrite the section of Introduction. In this section, the authors keyed point on the process of egg laying and mating of the cuttlefish, FLGamides or orcokinin B-like neuropeptides were not presented adequately. Please add more introduction about your aimed peptides.
Change made in the text.
- Section 2.1, FLGamide precursors were only identified by transcriptome information and this is not enough to confirm your results and followed experiments. The precursors should be cloned and identified by molecular experiments.
NGS and bioinformatics treatments have largely proven themselves for more than 10 years. Many transcriptomes have been published in specialized journals. The protein precursors, although incomplete, studied in this paper are therefore well established. Furthermore, the neuropeptides studied were all detected by mass spectrometry, which reinforces the sequences presented in this article.
- Section 2.3 Tissue mapping by immunocytochemistry was performed. But in Figure 3 and figure 4, the results showed that low expressions of FLGamides and PYY were observed. How to explain the neuropeptides function at low expression level?
Neurohormones act at extremely low physiological concentrations, so it is not surprising to observe weak immunological labeling. This low labeling is also the guarantee of a specific labeling. Moreover, the release of neurosecretions associated with reproduction are often released by pulse, which induces an almost tital emptying of endocrine cells.
- As we know than several mature peptides were predicted. Some did not show the functions what the authors expected. Whether the authors tried to use combined peptides to carry out the experiment? The results might be different. For example, FlGa1+FlGa3?
No, but it's a great idea. We are planning a study targeted exclusively on the functionality of orcokinins B, these are biological tests on the ovarian stroma under in vitro conditions, with a mixture of different neuropeptides, respecting the ratios indicated by the precursors, but also experiments in vivo by injection at the level of the vena cava into sexually mature females. These are extremely heavy experiments to implement, in particular on animals taken from the natural environment.
- If possible, I suggest the authors to carry out the experiments with two antibodies at the same treatment.
This approach using the 2 antibodies simultaneously was considered but ultimately rejected. The colocalization of the 2 types of orcokinins in the same secretion granules could have caused a masking phenomenon and a competition between the 2 antibodies potentially inducing non-specific labeling.
- Line 1299, how can the authors get the results that both neuropeptides had a dose -dependent activity on the mature distal oviduct?
I don’t see anything in the text about a dose – dependent activity of both neuropeptides FMRFamide and FLGamide.
Citation of lines 295-302 :
« Similar activity from serotonin of ovarian origin has already been described in cuttlefish [32] and by SepCRPs [2,3]. At the level of the main nidamental gland, which secretes the outer capsule of the egg, the results of the myotropic tests were more heterogeneous. However, thanks to the synergy test carried out in this study, we highlighted that FLGamide worked in association with FMRFamide, and increased its myoexcitator effect. Nevertheless, the effects of FLGamides alone showed that they are probably involved in the secretory mechanisms that release capsular products. »
- Section 2.5, biological activity, the positive or negative control should be added.
Each injection of synthetic peptide is preceded by the injection of perfusion liquid to which phenol red is added. This is not shown in the figures, but has been added in the text. However, I do not see what could be used as a positive control?
8) Line 360: Mention the body weight and shell length with mean ± SD.
Change made in the text.
9) Line 40, 109, 256…., Latin names should be italic.
Change made in the text.
10) Line 480, revise the format of the title.
Change made in the text.
11) Line 178, ‘are’ should be ‘is’
Change made in the text.
12) References: some of them lack of doi information and the Latin names in the references should also be ‘italic’.
Corrected and completed when it was possible.
For these two refrences, they are not DOI avalaible.
Boletzky, S. Fecundity variation in relation to intermittent or chronic spawning in the cuttlefish, Sepia officinalis (Mollusca, Cephalopoda). Bull. Mar. Sci. 1987, 40, 382–387.
Tompset, D.H. Sepia; The University Press of Liverpool: Liverpool, 1939.
Reviewer 3 Report
1) The manuscript is unacceptable in it’s current form because the disorganization where none of the section are related to each other.
2) The manuscript never addresses the stated aim: “... to elucidate the possible role of FLGamide in the regulation of egg-laying mechanisms in cuttlefish.”
The authors need to restructure their manuscript to provide a coherent order of information on the evidence for bioactivity.
3) Abstract: This should present and summarize the actual findings.
Introduction: Fails to justify the rest of the manuscript.
4) Results:
a) Section titles are repeated first ones are:
“2.1. Identification of FLGamide precursors” and “2.1. Animals and tissue collection”.
“2.2. Tissue mapping by mass spectrometry” and “2.2. In silico analyses”.
b) These sections are not connected to the Introduction and should not be presented.
“2.1. Identification of FLGamide precursors” is irrelevant because it was already reported by Zatylny-Gaudin et al. (2016). Rather the authors must describe how the sequences in Figure 2 were obtained.
“2.2. Tissue mapping by mass spectrometry”: No justification that the “main nidamental glands (MNGs)” are involved in “regulation of egg-laying mechanisms”.
“2.4. Structure determination”: What has this to do with “ regulation of egg-laying mechanisms”?
c) There are no results pertaining to the ‘role’ of FLGamide on regulation of egg-laying mechanisms. The only experimental data shows that FLGamide has bioactivity.
5) Please provide correct common name because there are ~100 known ‘cuttlefish’ species https://en.wikipedia.org/wiki/Sepia_(cephalopod)!
6) Where is the ‘English Channel’? Common cuttlefish (Sepia officinalis) is found in Mediterranean, North Sea, Baltic sea and North Atlantic - see for example: https://animaldiversity.org/accounts/Sepia_officinalis/
7) Use months not seasons as this is shows unacceptable Northern hemisphere bias.
8) Provide information on the neuropeptide ‘family’. Neuropeptides do not have ‘families’ because these are cleaved products of larger proteins from genes that have a ‘family’. Explain what “these neuropeptides” are found and what is a “have a hormone-like role” that is not found with neuropeptides.
9) Figure 1 A was previously published and could be considered a copyright violation. It is also irrelevant due to Figure 2. Figure 2 must highlight the ‘six neuropeptides’ (and all copies of these). Since neuropeptides are required to have confirmed bioactivity, it must be stated if these are experimental confirmed neuropeptides, predicted neuropeptides (homology to known neuropeptide) or predicted peptides (no reported bioactivity).
10) “2.2. In silico analyses” within 4. Materials and Methods does not describe the bioinformatics approached presented in the manuscript.
11) Figures 9 and 10 are results not Discussion and must be presented in the Material and Methods and Results section.
10) Multiple sequence alignment should involve the complete prohormone sequences and include other related species. In particular, species from the cephalopods such as Octopodiformes superorder, other mollusks, and other Metazoa species are required.
12) What experimental evidence to you have that orcokinins are not involved in "Feeding circuit activating neuropeptides"? Neuropeptides have been demonstrated to have multiple actions. Since there is only homology, the line 328-330 should be deleted.
13) The authors must utilize the Sepia pharaonis genome (BioProject PRJEB33343) to validate protein sequences from alternative splicing. Also to confirm the start and end of the sequences.
14) The authors need to connect the structure to the activity. It is useful but the authors need to identify the amidated Glycine and infer how the other similar predicted peptides may bioactive or not.
15) None of the mentioned transcriptomic and proteomic data have been provided as required by MDPI (https://www.mdpi.com/journal/marinedrugs/instructions#suppmaterials). The Data Sharing links to irrelevant data to the current study. Is this the same data previously used Zatylny-Gaudin et al. (2016)? That paper does refer transcriptomic and proteomic data where only the transcriptomic data is available (only by the BioProject identifier included here). Data should be reused but the authors need to be ensure that their actual results are their own and not results from a prior paper. For example, Figure 1C is too similar to Figure S-4(b) atylny-Gaudin et al. (2016).
Author Response
Dear reviewer,
please find attached a new version responding to constructive comments from reviewers

Reviewer 4 Report
This manuscript, by Endress, et al. is a follow-up to this group’s 2016 study of cuttlefish neuropeptides. In this manuscript, the authors focus on neuropeptides from a single prohormone (renamed “orcokinin B-like”) and perform localization, structural, and functional studies of the peptides from this prohormone. The research performed provides greater insight into the orcokinin B-like signaling system in cuttlefish, and physiological experiments are consistent with a role for these peptides in reproduction.
The manuscript is well-written and interesting. However, several things are unclear and critical data is missing. In order to substantiate the conclusions presented in this manuscript, a few changes are required:
Major changes
11) No MS data is shown, yet conclusions are being drawn from MS experiments. In order to draw conclusions, data must be shown to back up assertions.
22) References to “data not shown” and “unpublished data” should be removed. Conclusions should be drawn from data presented in the manuscript or previously published in peer-reviewed journals.
33) The purpose of the NMR experiments (especially in DPC micelles) and the conclusions drawn from this are not convincing. Just because a peptide can adopt a helical conformation in micelles does not say much about its structure in aqueous solution. In fact, most short peptides are expected to be unstructured in solution until receptor binding. The explanation that helical structure in DPC micelles suggests that the peptide first inserts into the membrane and then binds the receptor is a stretch.
44) The myotropic bioassay experiments lack positive and negative controls.
55) The data availability statement points to data gathered from a previous publication from this group (reference 21). It is thus not clear if all of this data is being reanalyzed from this previous publication, or if this is original data generated after the publication of this 2016 paper.
Minor changes
11) In Figure 1C: It is unclear exactly what is being measured (this is transcriptome data being reanalyzed, right?). More details on the figure or figure caption will help to clear this up.
22) Figure 1 caption: “profils” is a typo
33) Panel labels in Figure 8 are labeled A, C, D, do not match figure caption.
Author Response
Dear reviewer,
we thank you for your comments which helped us to improve our article.
Sincerely yours,
Joël Henry
Concerning
1) No MS data is shown, yet conclusions are being drawn from MS experiments. In order to draw conclusions, data must be shown to back up assertions.
Mass spectrometry data is detailed in a figure which will be provided as supplementary data.
2) References to “data not shown” and “unpublished data” should be removed. Conclusions should be drawn from data presented in the manuscript or previously published in peer-reviewed journals.
Change made in the text.
3) The purpose of the NMR experiments (especially in DPC micelles) and the conclusions drawn from this are not convincing. Just because a peptide can adopt a helical conformation in micelles does not say much about its structure in aqueous solution. In fact, most short peptides are expected to be unstructured in solution until receptor binding. The explanation that helical structure in DPC micelles suggests that the peptide first inserts into the membrane and then binds the receptor is a stretch.
We agree with the referee that such short peptide is expected to be unstructured in water as already written in the text. Accordingly, with the comment we introduce a direct binding to the receptor as the origin of helical conformation.
4) The myotropic bioassay experiments lack positive and negative controls
Each injection of synthetic peptide is preceded by the injection of perfusion liquid to which phenol red is added. This is not shown in the figures, but has been added in the text. However, I do not see what could be used as a positive control?
5) The data availability statement points to data gathered from a previous publication from this group (reference 21). It is thus not clear if all of this data is being reanalyzed from this previous publication, or if this is original data generated after the publication of this 2016 paper.
Data in Figure 1 is from Zatylny et al 2016. This figure is intended to provide an introduction to the article. It is also in the introduction and not in the section “results”.
Minor changes
- In Figure 1C: It is unclear exactly what is being measured (this is transcriptome data being reanalyzed, right?). More details on the figure or figure caption will help to clear this up.
Caption has been completed.
2) Figure 1 caption: “profils” is a typo
Change made in the figure.
3) Panel labels in Figure 8 are labeled A, C, D, do not match figure caption.
Change made in the figure.
Round 2
Reviewer 2 Report
I have seen the efforts that the authors revised the manuscript. I still insisted on explaining the question: V2 Line 212, how the authors can get the results that both neuropeptides had a dose -dependent activity on the mature distal oviduct?
This paper can be published after the explanation of the above question.
Author Response
Dear reviewer,
Thank you very much for your comments. Corrections have been made according to the your request concerning the dose dependent activity of the two neuropeptides in the figures 6A and 6B. Furthermore, I noticed an inversion of the 10-8 M and 10-6M concentrations in Figure 7 and the text relating thereto. Corrections have been made in the figure and in the text. sincerely yours, Joël henry
Reviewer 4 Report
I thank the authors for their careful attention to comments and critiques in their response. The changes to nearly all of my comments are appropriate and have resulted in an improved manuscript.
Author Response
Dear reviewer,
Thank you for your comments,
sincerely yours